# Computed Tomography Confirms Increased Left Atrial Volume in Patients with Bayés Syndrome Referred for Catheter Ablation of Atrial Fibrillation

**DOI:** 10.3390/diagnostics14212416

**Published:** 2024-10-30

**Authors:** Gabriel Cismaru, Gwendolyn Wagner, Gabriel Gusetu, Ioan-Alexandru Minciuna, Diana Irimie, Florina Fringu, Raluca Tomoaia, Horatiu Comsa, Bogdan Caloian, Dana Pop, Radu Ovidiu Rosu

**Affiliations:** Fifth Department of Internal Medicine, Cardiology Rehabilitation, “Iuliu Hatieganu” University of Medicine and Pharmacy, 400347 Cluj-Napoca, Romania

**Keywords:** left atrium, P wave, interatrial block, computed tomography

## Abstract

**Background:** Bayés syndrome is a recently identified condition that is defined by the presence of an interatrial block on a surface electrocardiogram, in addition to atrial arrhythmias such as atrial fibrillation, tachycardia, or left atrial flutter. This syndrome is linked to an increased risk of stroke, morbidity, and mortality. An interatrial block is a conduction delay between the right atrium and left atrium and can be recognized by a P wave duration >120 ms. It is known that P wave duration can estimate the size of the left atrium measured via echocardiography, which is a marker for stratifying cardiovascular risk. Our study aims to verify whether the duration of the P wave can estimate the volume of the left atrium measured by computed tomography in patients with an interatrial block. **Methods:** We included 105 patients with a sinus rhythm and a partial or advanced interatrial block (IAB) who underwent contrast-enhanced cardiac computed tomography (CT). The mean age was 62.2 ± 10.1 years, and 38% of the patients were women. **Results:** The mean P wave duration was 122.6 ± 11.4 ms in the partial IAB group and 150 ± 8.4 ms in the advanced IAB group (*p* < 0.01). The mean left atrial volume was 115 ± 39 mL in the partial IAB group and 142 ± 34 mL in the advanced IAB group (*p* = 0.001). P wave duration was longer in patients with an advanced as opposed to partial interatrial block. Left atrial volume and LAVI were higher in patients with an advanced as opposed to partial interatrial block. **Conclusions**: All the patients (100%) with an advanced IAB had a dilated left atrium. P wave duration can accurately estimate LA volume in patients with an IAB using the formula: LA volume = 0.6 × P wave + 46 mL.

## 1. Introduction

Bayés syndrome is characterized by an advanced interatrial block (IAB) and an increased risk of atrial arrhythmias such as atrial flutter, atrial fibrillation, and focal atrial tachycardia. An IAB is a delay in the conduction of the electrical impulse between the right atrium and left atrium due to a delayed or blocked conduction through the Bachman bundle [1,2]. It is one of the most valuable predictors for the recurrence of atrial fibrillation. Two types of IABs have been described: partial and advanced. In a partial IAB, the P wave is prolonged >120 ms, and in the advanced type, it is prolonged >120 ms with a bifid P (positive/negative) P wave. An advanced interatrial block in patients with an atrial fibrillation ablation is predictive of a recurrent post-ablation [3]. Additionally, atrial fibrosis results in structural remodeling, blood stasis, and thrombus formation, which implies that an IAB is linked to an elevated risk of stroke and other peripheral embolic events [3].

On an electrocardiogram, the P wave indicates atrial depolarization and is regarded as an accurate measure of atrial size and atrial conduction time. Changes in the atrial refractory period and conduction velocity can serve as the basis for reentry and triggered activity, leading to the occurrence of atrial fibrillation [4]. Among a group of 285,933 individuals, a prolonged P wave duration (120–129 ms) and very prolonged P wave duration beyond 130 ms were associated with an increased risk of developing atrial fibrillation [5].

It is known that the duration of the P wave can estimate the size of the left atrium expressed as its diameter when measured via echocardiography [6,7]. Enlargement of the left atrium is suggestive of a risk of future atrial fibrillation, stroke, heart failure, and death [8]. The volume of the left atrial has also been recognized as an indicator of the process of atrial remodeling. While there exists a substantial amount of data regarding the correlation between the LA diameter and the duration of the P wave, there is a paucity of data regarding the correlation between LA volume and P wave duration. Prior research has demonstrated that the estimation of 2-dimensional LA volume using diameter underestimates the actual LA volume by an average of 20% [9,10]. By virtue of its tendency to adopt a more spherical shape during enlargement and remodeling, the volume of the LA can be regarded as a more sensitive indicator of LA remodeling compared to a simple LA diameter measurement.

Cardiac multidetector computed tomography has been introduced as a promising modality for coronary artery imaging. Its application in the quantification of chamber volume, including the left atrium, has been demonstrated through other research [11,12]. Our study aims to verify whether the duration of the P wave can estimate the volume of the left atrium measured by computed tomography.

## 2. Materials and Methods

### 2.1. Study Population

In this retrospective study, we included 105 patients with sinus rhythm and a partial or advanced interatrial block who underwent contrast-enhanced cardiac CT in a tertiary-care-teaching hospital. All patients signed an informed consent, and the study was approved by the hospital’s ethics council. Clinical investigations were conducted according to the Declaration of Helsinki, in line with the guidelines for good clinical practice. Age, sex, height, weight, BMI, and body surface area (according to the Mosteller formula [2] were recorded for all subjects. 

The inclusion criteria were as follows: patients aged 18 to 80 years, diagnosed with paroxysmal or persistent atrial fibrillation unresponsive to antiarrhythmic therapy, who provided consent for the ablation intervention.

The exclusion criteria were as follows: patients under 18 years of age or over 80 years of age, left atrial diameter >55 mm, presence of thrombus in the left atrium or left atrial appendage, NYHA class IV heart failure or cardiogenic shock, hemiplegia, or incapacity to provide consent for the invasive procedure. As all the patients underwent CT scans, we excluded those with creatinine >1.2 mg/dL, an allergy to iodinated contrast medium, or pregnancy.

### 2.2. ECG and CT Analysis

For the detection of a presence or absence of an interatrial block, a 12-lead ECG was analyzed at a paper speed of 25 mm/sec with an amplitude of 10 mm/mV. The P wave duration measurement was performed by an investigator (GW) who was blinded to the clinical status of the patient. The onset of the P wave was defined as the junction between the isoelectric line and the beginning of the P wave. The P wave offset was defined as the junction between the end of the P wave and the isoelectric line. Patients were divided into 2 groups: those with a partial interatrial block (defined as a P wave duration of over 120 ms) and those with an advanced IAB (P wave duration over 120 ms with a biphasic positive/negative morphology). All patients participating in the study were admitted for an atrial fibrillation catheter ablation. Before the procedure, non-ECG-gated computed tomography angiography was performed in all 105 patients with the aim of checking the anatomy and number of pulmonary veins as well as the presence or absence of thrombus in the auricle of the left atrium. A three-dimensional reconstruction was used for volume calculation, with the left atrial appendage and pulmonary veins being excluded from the LA volume measurement. 

### 2.3. Statistical Analysis

All data collected from patients were entered into an Excel file, which was then exported to an SPSS file. Continuous data were described using the mean and standard and median deviation, and for ordinal or nominal data, frequencies and percentages were used. For the association between P wave duration and the volume of the left atrium, a Pearson correlation (or Spearman’s in case of a non-normal distribution) was used, and for the establishment of the volume calculation formula according to P wave duration, a linear regression was used. The comparison between the estimated and actual volume measured via CT was made using Bland–Altmann graphs of the differences and means between the two evaluations. SPSS version 21 was used for all the statistical analyses, considering a significant value of *p* below 0.05.

## 3. Results

Our study group included 105 patients (mean age 62.2 ± 10.1 years, 38% women), all with an interatrial block. Of these, 81 had a partial interatrial block, and 24 had an advanced interatrial block. The baseline characteristics are summarized in Table 1. The duration of the P wave was longer in patients with an advanced as opposed to partial IAB (150 ± 8.4 ms vs. 122.6 ± 11.4 ms; *p* < 0.001). Eighty-four patients (80%) had a left atrial volume >90 mL, and 21 (20%) had a volume <90 mL. According to the LAVI, 89 patients (84.8%) had a dilated left atrium (>40 mL/m^2^), and 16 patients (15.2%) had a normal left atrium. 

The LAVI was also higher in the advanced IAB group compared to the partial IAB group (66.5 mL/m^2^ vs. 54.1 mL/m^2^ *p* = 0.003).

All the patients with an advanced interatrial block had a dilated left atrium: 100% had an AS volume >90 mL, and 100% had a LAVI > 40 mL/m^2^.

The duration of the P wave correlated with the left atrial volume, the strongest associations being found in female patients and those aged >70 years (Table 2).

Using linear regression, we determined a formula to estimate the left atrial volume based on the duration of the P wave (Figure 1):LAVI (mL) = 0.34 × P wave + 8 mLAS (mL) = 0.6 × P wave + 46 mL

To determine if the estimation formula was accurate, we performed a Bland–Altmann analysis [4] and plotted on a graph the differences and means between the estimated and measured LA volumes. The resulting graph is an XY scatter graph, in which the Y axis shows the difference between the estimated volumes using the formula and the measured volumes using computed tomography, and the X axis represents the average of these measurements ((estimated + measured)/2). In other words, the difference between the estimated and measured volumes is plotted against the average of the two volumes. We obtained the graph that can be seen in Figure 2, with the majority of the points situated within ±2 s of the mean difference, with a bias of 4.87 and limits of agreement [2.85–12.6].

## 4. Discussion

The size of the left atrium is a marker for stratifying cardiovascular risk [13,14,15]. Dilatation of the left atrium is a predictor not only of atrial fibrillation but also of coronary artery disease, congestive heart failure, myocardial infarction, stroke, and death [16,17,18,19]. An interatrial block is characterized by the presence of fibrosis in the Bachmann’s bundle, extending from both the right atrium and left atrium. An assessment of the extent of atrial fibrosis using echocardiography and cardiac magnetic resonance imaging enables the evaluation of the atrial tissue in patients with an IAB and Bayés syndrome [20]. When fibrosis affects both the right and left atrium, it also impairs the transmission of electrical signals from the sinus node to the Bachmann’s bundle and subsequently to the AV node [21]. This results in a prolongation of the P wave and the distinctive morphology of an interatrial block. 

The size of the left atrium can be evaluated using echocardiography and computed tomography. The anteroposterior diameter in echocardiography is inadequate for an accurate estimation of the left atrium’s dimensions. More suitable alternatives include calculation of the volume using the ellipsoid or Simpson method [22]. Considering the excellent capacity of computed tomography to assess both the diameter and volume of the left atrium, we opted to adopt this technique in our study.

Sometimes, it may be convenient to use ECG to predict left atrial volume. In our study, we described a formula for estimating left atrial volume, based on the P wave duration of a 12-lead electrocardiogram. The P wave measurement is a rapid step in the ECG reading process, and our formula can predict with good accuracy both LA volume and LAVI.

The Bachmann bundle is the predominant interatrial conduction pathway. However, the conduction of electrical impulses between the right atrium and left atrium is facilitated by other pathways such as the fossa ovalis and coronary sinus. Despite the presence of interatrial tracts in the Bachmann bundle, the fossa ovalis and coronary sinus themselves play a role in interatrial conduction [23]. This is supported by the observation made by Lemery et al., who found that the right and left parts of the atrial septum were activated separately and asynchronously from one other [24]. A similar mechanism applies to the right-sided ostium and the left body of the coronary sinus. 

Previous studies have shown that the atrial chambers are activated sequentially and contribute equally to the generation of the P wave. For instance, the mid-third of the P wave indicates equal contributions from both the right and left atria, while the first and last thirds of the P wave represent the right atrium and left atrium, respectively [25].

Research has demonstrated that the presence of biphasic P waves in the inferior leads is a reliable indicator of a block in the Bachmann bundle. Holmqvist et al. showed through electroanatomical mapping that the atrial activation direction was both superior-to-inferior-to-superior and posterior-to-anterior-to-posterior. This pattern arises from conduction mostly through the coronary sinus, without any additional conduction occurring through the Bachmann bundle or the fossa ovalis. The findings of Holmqvist et al. are significant as they provide a straightforward approach to determine interatrial conduction using only the surface ECG and the likelihood of atrial arrhythmias when conduction occurs exclusively through the coronary sinus [26]. 

An interatrial block is a conduction disorder of the electrical activity between the right and left atrium and results in a prolonged P wave. There are two types of IABs: partial and advanced. In an advanced IAB, activation through the Bachmann’s bundle, which conducts electrical impulses from the right atrium to the left atrium, is completely blocked, and the P wave has a “plus/minus” morphology as atrial activation takes place retrogradely from the inferior wall near the AV junction to the upper atrial part [1]. The association between an advanced interatrial block and atrial tachyarrhythmias is known in the medical literature as Bayés syndrome [27,28]. There are numerous studies that demonstrate the predictive capacity of an interatrial block: for the development of atrial fibrillation, for recurrences of atrial fibrillation after electrical conversion or radiofrequency ablation [28,29,30,31,32,33]. An interatrial block is also an indirect sign of fibrosis in the Bachmann bundle. Fibrosis that extends to the left atrium may explain the higher recurrences of AF after radiofrequency ablation in patients with an interatrial block [33]. Our study shows that in patients with an interatrial block, LA volume can be estimated using the following formula: LA volume = 0.6 × P wave + 46 mL.

Our results are in line with those of Ariyarajah et al. [6], who estimated LA anteroposterior diameter based on P wave duration using the following formula: LA = 0.29 × P wave + 2.47. They studied a sample of 66 individuals, of whom 33 had an IAB and 28 did not. P wave duration was measured using a standard 12-lead ECG at a paper speed of 25 mm/s and an amplitude of 10 mm/mV. LA size was measured as the anteroposterior diameter in the parasternal long axis view. Mean LA diameter was 36.7 mm in the control group and 42.3 mm in the IAB group (*p* = 0.004). LA size could be calculated using the aforementioned formula. However, there are major differences between our methodology and Spodick’s: we determined the dimensions of the left atrium by measuring its volume as evaluated via computed tomography; while they used the anteroposterior diameter measured via bidimensional echocardiography. The volume is much more accurate than a single diameter for LA estimation; therefore, we expressed both LA volume and LAVI based on P wave duration. The second important difference is the number of patients included: there were 38 in Spodick’s study and 105 in our group. Lastly, but not least, we differentiated patients into two categories of interatrial blocks, partial and advanced, based on P wave duration and morphology, and we presented our findings in accordance with this grouping.

Based on the study of Taina et al. [34], the upper threshold value for normal left atrial volume was estimated at 90.4 mL. They studied 146 patients with a suspected stroke of cardiac origin and compared the values with a control group of 40 healthy individuals. A comparative analysis of left atrial volume was conducted using computed tomography and transthoracic echocardiography. In the controls, the mean LA volume was 59.8 mL, and the mean LA diameter was 30.4 mm. Based on these values, they calculated the upper limit for normal LA dimensions using the mean values plus twice the standard deviation: 90.4 mL for LA volume and 40.4 mm for LA diameter. 

Stojanovska et al. [35] determined the normal values for LA volume and LAVI, both in men (86 mL and 41 mL/m^2^) and women (74 mL and 40 mL/m^2^), respectively). They studied 74 patients with a chest pain of non-cardiac cause and normal coronary CT angiography and measured LA volume and function on a 64 multi-detector CT scanner. LA diameter was measured in an anterior–posterior direction parallel to LVOT, and LA volume was measured at its maximum size and afterwards indexed to the body surface area. The normal reference values for the left atrium volume were determined by the authors using age and sex as parameters. Nevertheless, they noted that age did not affect LA volume after adjusting for sex and body surface area. Consequently, the authors suggested that LA volume should be indexed to body surface area. The authors concluded that the LA volume measurement is a more precise and reliable estimation of LA enlargement than LA anteroposterior diameter, as LA enlargement is not symmetric in all directions, i.e., anteroposterior, laterolateral, and superoinferior.

In the ROMICAT study [36], 377 patients with chest pain who presented to the emergency department received 64-slice computed tomography to assess their coronary arteries. The control group comprised individuals who did not have plaques, as identified by CT angiography, and were free from risk factors such as dyslipidemia, hypertension, and diabetes. Based on the ROMICAT classification, which considers a normal left atrial volume of 90 mL via computed tomography and a left atrial volume index of 47 mL, 80% of our patients with an interatrial block had an increased left atrial volume, and 85% had an increased left atrial volume index (LAVI). In all of our patients with an advanced intra-aortic valve (IAB) and 75% of patients with a partial IAB, the left atrium was found to be dilated, exceeding 90 mL. 

Our study included patients with an interatrial block and atrial fibrillation, known as Bayés syndrome. However, in healthy individuals, the relationship between P wave duration and LA dimensions does not apply. Petersson et al. studied 504 healthy male professional soccer players that had a median age of 25 years. All of them underwent echocardiography and a 12-lead ECG. P wave duration had a mean of 132 ms, and the LA diameter was 35 mm, with an indexed diameter of 17 mm/mp, a volume of 71.2 mL, and an indexed volume of 35.8 mL/mp [37]. Therefore, our formula applies to patients with Bayés syndrome. 

In clinical practice, it is often necessary to compare a new method with a standard of measurement. We compared volumes obtained with our formula to volumes obtained by direct measurement via computed tomography, using Bland–Altmann plots of difference between the estimated and observed values. In 1983, Altman and Bland proposed an analysis based on quantifying the agreement between two quantitative measurements by studying the mean difference between them and constructing the limits of the agreement. These statistical limits are calculated using the mean and standard deviation of the differences between two measurements [38]. A very good overlap was obtained between the volumes estimated by our formula and the volumes measured via computed tomography [bias = 4.87, limits of agreement: 2.85–12.5], demonstrating that the estimation formula is accurate and can be used in clinical practice: LA volume = 0.6 × P wave + 46 mL. 

### Limitations

This study was carried out in a single center; therefore, prospective multicentric studies are required for larger populations. A cardiac MRI is considered the standard for measuring cardiac chamber volumes, but for economic reasons, we could not perform an MRI for all patients. All our patients (100%) with an advanced IAB and 75% of the patients with a partial IAB had a dilated left atrium (>90 mL). Therefore, our formula for estimating left atrial volume applies to prolonged P waves and mostly dilated atria. Probably for patients with normal atrial volumes and P waves below 120 ms, the calculation formula would be different, which should be demonstrated in a future study.

## 5. Conclusions

Left atrial volume and LAVI were higher in patients with an advanced as opposed to partial interatrial block. All patients with an advanced IAB had an atrial volume > 90 mL and a LAVI > 40 mL/m^2^. P wave duration was longer in patients with an advanced as opposed to partial interatrial block. P wave duration is a simple and quick measurement that can accurately estimate LA volume in patients with an interatrial block using the following formula: LA volume = 0.6 × P wave + 46 mL and LAVI = 0.4 × P wave + 8 mL. Being inexpensive, rapid, and easily obtained, the ECG detection of an interatrial block could offer an appropriate method to detect patients with a left atrial volume that is a risk for developing atrial fibrillation.

## Figures and Tables

**Figure 1 diagnostics-14-02416-f001:**
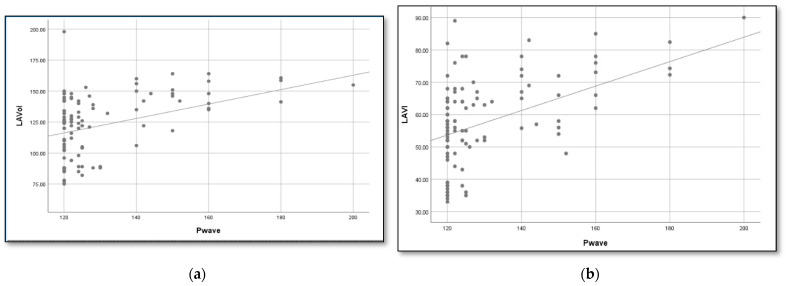
Association between P wave duration and LA volume in patients with an IAB. According to the linear regression, the formulas that best estimate the LA volume and index are (**a**) LA volume = 0.6 × P wave + 46 mL and (**b**) LAVI = 0.4 × P wave+ 8 mL.

**Figure 2 diagnostics-14-02416-f002:**
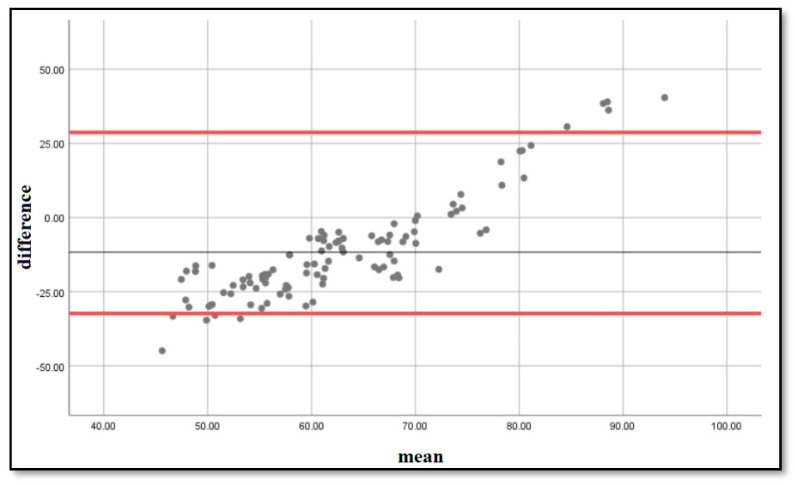
Bland–Altmann plots of the differences and means of the measured and estimated LA volumes. The 5% and 95% percentiles are marked with a red line, and the mean difference is marked with a black line.

**Table 1 diagnostics-14-02416-t001:** Baseline characteristics of the study group divided into two categories: partial and advanced IAB.

	Partial IAB *n* = 81	Advanced IAB *n* = 24	*p* Value
Age	63.2 ± 10.1	63.3 ± 10.7	0.957
Sex (female No.,%)	38 (36%)	(23) 22%	0.298
Weight (kg)	65 ± 13.3	75 ± 9.9	0.776
Height (m)	162 ± 6.1	160 ± 6.6	0.413
BMI (kg/m^2^)	21.5 ± 3.7	24.5 ± 2.5	0.428
BSA (m^2^)	1.75 ± 0.1	1.81 ± 0.1	0.924
LA volume	115.1 ± 38.9	142.1 ± 34.3	0.003
LAVI mL/m^2^	54.1 ± 18.3	66.5 ± 14.8	**0.003**
LA volume > 90 mL	60 (74%)	24 (100%)	**0.001**
LAVI > 40 mL/m^2^	65 (80.2)	24 (100%)	**0.001**
P wave duration	122.6 ± 11.4	150 ± 8.4	**0.0001**

**Table 2 diagnostics-14-02416-t002:** Correlation coefficients between P wave duration and left atrial volume for six patient categories.

Category	R Coefficient of Correlation	*p* Value
Male gender	0.531	<0.001
Female gender	0.768	<0.001
Obese	0.604	<0.00001
Non-obese	0.552	<0.01
Older (age > 70) Age < 70	0.687 0.485	<0.0001 <0.0001

## Data Availability

Data can be found on the mega cloud repository https://mega.nz/fm/0F0yGIDT.

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
