# Peer review of "Computed Tomography Confirms Increased Left Atrial Volume in Patients with Bayés Syndrome Referred for Catheter Ablation of Atrial Fibrillation"

_diagnostics, 2024, doi:10.3390/diagnostics14212416_

Round 1

Reviewer 1 Report

Comments and Suggestions for Authors

I thank the Editor for the opportunity to review the manuscript.

I congratulate the authors on a clear work.

I will not comment on the topicality of the topic because I am not qualified to do so. From the point of view of the topicality of the topic, I leave the inclusion of the manuscript in the journal to the editor.

I have a few questions about the methodology and results:

The authors present the work as retrospective. Thus, the authors obtained data only from already available medical documentation. For what indication was the patient subjected to a CT scan? For what reason were the patients hospitalized? How was the selection of patients? What were the inclusion and exclusion criteria?

The authors state that only patients with IAB (line 71) were included in the study. IAB is defined as P > 120ms (line 38). It is obvious from Figure 1 that the group of patients also included patients with P<120ms. What's more, even patients with p<100ms. I would like to ask the authors to correct or clarify this inconsistency.

Author Response

Gsbriel Cismaru, MD, PhD

“Iuliu HaÅ£ieganu” University of Medicine and Pharmacy

Viilor 46-50 street,  Cluj-Napoca, 400347, Romania

gabi_cismaru@yahoo.com

October 22 , 2024

Dear Editor and Reviewers,

Thank you for your comments and suggestions. You raised important issues, and the inputs are very helpful for improving the manuscript. We agree with  almost all the comments, and we have revised our manuscript accordingly.

 Please find below the point-by-point response to each of the raised questions. Please also  find attached the revised version of the manuscript and changes highlighted with red color.

Response to Reviewer 1.

I thank the Editor for the opportunity to review the manuscript. I congratulate the authors on a clear work. I will not comment on the topicality of the topic because I am not qualified to do so. From the point of view of the topicality of the topic, I leave the inclusion of the manuscript in the journal to the editor. I have a few questions about the methodology and results:

Comment 1: The authors present the work as retrospective. Thus, the authors obtained data only from already available medical documentation. For what indication was the patient subjected to a CT scan? For what reason were the patients hospitalized? How was the selection of patients? What were the inclusion and exclusion criteria ?

Response 1 : The reviewer is correct. The motivation for computed tomography, hospitalization, and the related inclusion and exclusion criteria were not properly articulated in the Methodology section of our study.

All patients included in the study were hospitalized in order to perform atrial fibrillation ablation. that's why pre-procedurally all of them had a CT scan with the aim of checking the anatomy and number of pulmonary veins, abnormal structure and number of veins as well as the presence or absence of thrombus in the auricle of the left atrium.

The inclusion criteria consisted of: patients aged 18 to 80 years, diagnosed with paroxysmal or persistent atrial fibrillation unresponsive to antiarrhythmic therapy, who provided consent for the ablation intervention.

The exclusion criteria included:  patients under 18 years of age, or over 80 years of age, left atrial diameter > 55 mm, presence of thrombus in the left atrium or left atrial appendage, NYHA class IV heart failure or cardiogenic shock, hemiplegia, or incapacity to provide consent for the invasive procedure. As all the patients underwent CT scans, we also excluded those with creatinine > 1.2 mg/dl, allergy to iodinated contrast medium or pregnancy.

Comment 2: The authors state that only patients with IAB (line 71) were included in the study. IAB is defined as P > 120ms (line 38). It is obvious from Figure 1 that the group of patients also included patients with P<120ms. What's more, even patients with p<100ms. I would like to ask the authors to correct or clarify this inconsistency.

Response 2: We regret the significant error in figures 1a and 1b and express our gratitude to the reviewer for bringing it to our attention. The figures in the image were inaccurate, as the study exclusively comprised patients with partial or advanced interatrial block. We corrected the data and incorporated the accurate graphs generated by SPSS.

        (a)                                                                    (b)

Figure 1: Association between P wave duration and LA volume in patients with IAB. According to linear regression, the formula that best estimates LA volume index is  (a) LA volume =0.6 x P wave + 46 ml and (b)LAVI=0.4 x P wave+ 8 ml.

My co-authors and I would be happy to answer any additional questions, should they arise. We thank you for your time and consideration and for giving us the opportunity to improve our manuscript. We hope that you will find our revised manuscript suitable for publication in Cardiology.

Sincerely,

On behalf of the Authors,

Dr. Gabriel Cismaru

Reviewer 2 Report

Comments and Suggestions for Authors

This is an interesting topic about a recently discovered syndrome and a useful study of diagnostic procedures. Bayes syndrome is defined as the presence of atrial block (IAB) on the ECG in addition to atrial arrhythmia. The syndrome is associated with an increased risk of morbidity and mortality, especially stroke. IAB is the conduction delay between the right and left atrium (LA) and can be identified by P wave duration >120 ms. This duration can estimate the size of LA measured by echo (which is a marker of cardiovascular risk). The study aimed to verify whether the duration of P-waves could estimate LA volume measured by synthetic tomography (CT) in IAB patients.

The trial included 105 patients (62.2+/-10.1 years old, 38% women) with partial or advanced IAB sinus rhythm receiving CT. The mean P-wave duration was 122.6 +/- 11.4 ms in the partial IAB group and 150 +/- 8.4 ms in the late IAB group (p<0.01). The mean LA volume was 115 +/ -39 ml in the partial IAB group and 142 +/ -34 ml in the advanced IAB group (p = 0.001). In advanced IAB, the P-wave duration is longer, the LA volume and LAVI are higher, and all patients have dilated LA. P-wave duration can be used to accurately estimate LA volume in IAB patients using appropriate formulas.

CT confirmed increased LA volume in Bayes syndrome patients referred for atrial fibrillation catheter ablation.

In short, the study was of high quality. The introduction, methods, and research are adequately described, the results are briefly presented, and the conclusions support the results. I recommend accepting the study.

Author Response

Gsbriel Cismaru, MD, PhD

“Iuliu HaÅ£ieganu” University of Medicine and Pharmacy

Viilor 46-50 street,  Cluj-Napoca, 400347, Romania

gabi_cismaru@yahoo.com

October 22 , 2024

Dear Editor and Reviewers,

Thank you for your comments and suggestions. You raised important issues, and the inputs are very helpful for improving the manuscript. We agree with  almost all the comments, and we have revised our manuscript accordingly.

 Please find below the point-by-point response to each of the raised questions. Please also  find attached the revised version of the manuscript and changes highlighted with red color.

Response to Reviewer 2.

Comment 1: This is an interesting topic about a recently discovered syndrome and a useful study of diagnostic procedures. Bayes syndrome is defined as the presence of atrial block (IAB) on the ECG in addition to atrial arrhythmia. The syndrome is associated with an increased risk of morbidity and mortality, especially stroke. IAB is the conduction delay between the right and left atrium (LA) and can be identified by P wave duration >120 ms. This duration can estimate the size of LA measured by echo (which is a marker of cardiovascular risk). The study aimed to verify whether the duration of P-waves could estimate LA volume measured by synthetic tomography (CT) in IAB patients.

The trial included 105 patients (62.2+/-10.1 years old, 38% women) with partial or advanced IAB sinus rhythm receiving CT. The mean P-wave duration was 122.6 +/- 11.4 ms in the partial IAB group and 150 +/- 8.4 ms in the late IAB group (p<0.01). The mean LA volume was 115 +/ -39 ml in the partial IAB group and 142 +/ -34 ml in the advanced IAB group (p = 0.001). In advanced IAB, the P-wave duration is longer, the LA volume and LAVI are higher, and all patients have dilated LA. P-wave duration can be used to accurately estimate LA volume in IAB patients using appropriate formulas.

CT confirmed increased LA volume in Bayes syndrome patients referred for atrial fibrillation catheter ablation.

In short, the study was of high quality. The introduction, methods, and research are adequately described, the results are briefly presented, and the conclusions support the results. I recommend accepting the study.

Response 1:  Thank you for your appreciation. You comprehended the core of our research and appreciated our findings, for which we extend our gratitude.

My co-authors and I would be happy to answer any additional questions, should they arise. We thank you for your time and consideration and for giving us the opportunity to improve our manuscript. We hope that you will find our revised manuscript suitable for publication in Cardiology.

Sincerely,

On behalf of the Authors,

Dr. Gabriel Cismaru

Round 2

Reviewer 1 Report

Comments and Suggestions for Authors

The authors explained both of my comments. I have no further comments on the methodology of the work. I recommend accepting the work.